# Genome-wide association studies of ischemic stroke based on interpretable machine learning



Stefan Nikolić[1], Dmitry I. Ignatov[1], Gennady V. Khvorykh[2],
Svetlana A. Limborska[2] and Andrey V. Khrunin[2]

[1] Laboratory for Models and Methods of Computational Pragmatics; Department of Data Analysis
   and Artificial Intelligence, HSE University, Moscow, Russia
[2] National Research Centre "Kurchatov Institute", Moscow, Russia

## ABSTRACT

Despite the identification of several dozen genetic loci associated with ischemic stroke (IS), the genetic bases of this disease remain largely unexplored. In this research we present the results of genome-wide association studies (GWAS) based on classical statistical testing and machine learning algorithms (logistic regression, gradient boosting on decision trees, and tabular deep learning model TabNet). To build a consensus on the results obtained by different techniques, the Pareto-Optimal solution was proposed and applied. These methods were applied to real genotypic data of sick and healthy individuals of European ancestry obtained from the Database of Genotypes and Phenotypes (5,581 individuals, 883,749 single nucleotide polymorphisms). Finally, 131 genes were identified as candidates for association with the onset of IS. *UBQLN1*, *TRPS1*, and *MUSK* were previously described as associated with the course of IS in model animals. *ACOT11* taking part in metabolism of fatty acids was shown for the first time to be associated with IS. The identified genes were compared with genes from the Illuminating Druggable Genome project. The product of *GPR26* representing the G-coupled protein receptor can be considered as a therapeutic target for stroke prevention. The approaches presented in this research can be used to reprocess GWAS datasets from other diseases.

## INTRODUCTION

### Background

Stroke is a medical condition caused by the interruption of the blood supply to a part of the brain, leading to cell death and subsequent dysfunction of brain tissue in that area. In general, stroke is the second or third most common cause of death and disability (*Katan & Luft, 2018*). Ischemic stroke (IS) comprises about 80% of all stroke events. Except for a small percentage of monogenic forms, most cases of IS are caused by a combination of environmental and genetic factors. The latter are represented by hereditary variations in DNA sequences, for example single nucleotide polymorphisms (SNPs), which are estimated to account for up to 40% of the risk of stroke (*Bevan et al., 2012*). To date, several

Corresponding authors
Dmitry I. Ignatov, dignatov@hse.ru
Andrey V. Khrunin,
khrunin-img@yandex.ru

tens of genetic loci have been robustly associated with IS. Their characterization allowed researchers to describe the underlying biological mechanisms of the disease and find potential drug targets for its treatment and prevention (*Debette & Markus, 2022*). One more important option is their use to quantify the genetic predisposition to the risk of IS (*i.e.*, identification of individuals at high risk of IS) by combining the effects of multiple significant genetic variants into a single score (polygenic risk score, PRS). The results of several studies demonstrated the efficacy of PRS in IS prediction. The larger the set of SNPs, the stronger the PRS performance was observed (*Rutten-Jacobs et al., 2018*; *Marston et al., 2021*; *Mishra et al., 2022*). However, the clinical utility of PRS in general remains low (*Koch et al., 2023*). One reason is that only a small part of the phenotypic variability can be explained with these genetic variations, highlighting the potential for the discovery of additional causative loci (genes) in the future.

The most loci associated with diseases were discovered in genome-wide association studies (GWAS), based mainly on statistical testing of the distribution of allele and genotype frequencies of individual SNPs in groups of subjects with a disease (cases) and without it (controls) (*Uffelmann et al., 2021*). This approach has some limitations, the essential of which is the inability to identify loci with small effect sizes and to account for the polygenic nature of common diseases (*Lappalainen & MacArthur, 2021*). The latter can be attributed with model-based algorithms processing SNPs simultaneously. Among such algorithms was logistic regression (LR) historically applied first to GWAS datasets (*Nicholls et al., 2020*). Finally, it became an alternative way to detect the susceptibility loci of IS as well (*Malik et al., 2018*; *Chung et al., 2019*). LR is known to be the most popular algorithm of supervised machine learning (ML) with tabular data. Therefore, other ML algorithms can be potentially utilized.

GWAS datasets are naturally suitable for processing with ML. Generally, the genotypic data can be given as a matrix where rows are samples (individuals), columns are SNPs, and values are genotypes. It is the object-attribute matrix, which is a standard way to represent the information in data mining, an interdisciplinary field of computer science and statistics. There are also given the labels denoting the belonging of sample to case or control groups. Altogether, one gets the input data for ML.

The processing of GWAS datasets based on ML consists of two essential steps. Firstly, the model predicting the binary outcome (case/control) is built with the use of algorithms of supervised ML. Secondly, the SNPs are ranked according to their influence on the outcome and those having the strongest relationship are selected. The usefulness of such approach has been demonstrated in different studies, some of them are mentioned below.

For example, preconditioned random forest (RF) was applied to predict the risk of developing radiation-associated contralateral breast cancer (RCBC) (*Lee et al., 2020*). The authors used 712 out of almost 800 K SNPs available from genotyping with DNA-microarray. They subsetted them by *p*-value of chi-square tests less than 0.001. The SNPs were ranked by variable importance measure. The RF modeling with selected SNPs resulted in higher AUC = 0.62 and allowed identifying more biologically relevant biomarkers, including those suggested in previous studies. The authors also checked the

possibility to apply the clinical variables, but none of them reached nominal significance ($p$-value < 0.05).

Sometimes, the simultaneous application of genetic and non-genetic variables in ML can improve the prediction quality. Several ML models to predict the anterior cruciate ligament (ACL) rupture in the labrador retriever dog were explored (*Baker et al., 2020*). The condition in dogs has a similar clinical presentation and progression as in humans. The best classifier was weighted RF that showed AUC of 0.792 (SD = 0.027) with five SNPs and three non-genetic risk factors (weight, sex, and neuter status). When no SNPs were removed for linkage disequilibrium (LD) and covariates were not considered, the best classifier was gradient boosted trees (AUC 0.59 (SD = 0.049), 10,000 SNPs). The researchers demonstrated the ability to predict ACL rapture in dogs. This work showed that ML will lead to advancements in both veterinary and human medical research.

Below, we showed that the processing of GWAS data with ML can be followed by statistical testing. Two approaches can be joined within one workflow in a more complex way. So did the authors searched for loci associated with hepatitis B surface antigen (HBsAg) seroclearance in a population of Korean patients with chronic hepatitis B (*Silva et al., 2022*). Firstly, they ranked all the SNPs using RF algorithm. After that, they compiled the top SNPs from RF and significant SNPs obtained in GWAS by statistical tests. Then, the hierarchical agglomerative clustering was performed. Finally, the resulting clusters of SNPs were tested for association with the phenotype using the Hamming distance-based association test. The integrated approach identified three susceptibility loci for HBsAg seroclearance: rs2399971, gene LINC00578, and locus 11p15, the first of which had been previously described in the literature as a biomarker for susceptibility whereas two others had been linked with cancer diseases associated with hepatitis B virus infection.

The common diseases including IS are considered to be associated with a set of casual polymorphisms, each contributing a minor effect on the phenotype (*Price, Spencer & Donnelly, 2015*). The ML methods operating on sets of variables sounds to be more appropriate for accounting for the polygenic nature of these diseases than statistical testing of individual SNPs. Some authors believe the ML methods will substitute totally the classical GWAS approaches in the future (*Nicholls et al., 2020*). At the same time, they also noted that although ML methods demonstrated good applicability for prioritization of GWAS results of particular diseases, they have also some limitations. We consider them can be minimized by combination of classical GWAS and ML methods that complement each other. In addition to several widely used ML methods (logistic regression and gradient boosting on decision trees) we also applied the interpretable neural network TabNet to rank SNPs. To explore all the results, we proposed Pareto optimality, a multi-objective optimization framework that has not been extensively explored in GWAS.

One of the widely discussed topics in GWAS is the interpretation of the results obtained. The annotation of selected genetic variants and gene over-representation analysis became a routine post-GWAS procedures to gain insights about possible biological background. To add, we propose to estimate the potential therapeutic value of genes identified. To start, they can be compared with genes of the Illuminating the Druggable Genome (IDG) project (druggablegenome.net). It includes 323 genes, which products are non-olfactory G-protein

coupled receptors (GPCR, 44%), ion channels (20%), and protein kinases (36%). These are the three most common drug-targeted protein families. The goal of IDG project is to study the properties and functions of those proteins that are currently not well studied within these families. Although our study focuses on the genetic bases of the onset of IS, rather than the course or outcome of the disease, we believe the common genes for different stages of pathogenesis may exist. Therefore, IDG genes were considered in this research as well.

## Contribution

In this study we conducted a genome-wide association study (GWAS) using chi-square testing of individual SNPs and applied three classification algorithms: logistic regression, gradient boosting on decision trees (GBDTs), and the interpretable neural network TabNet to rank SNPs based on their contribution to the outcome. Additionally, we introduced the concept of Pareto optimality to enhance the robustness of SNP selection by integrating multiple metrics.

Our approach differs from existing literature by incorporating a multi-metric evaluation, combining $p$-values from chi-square tests, logistic regression coefficients, TabNet attention scores, and GBDTs importance scores. This integrated methodology offers a more comprehensive evaluation of SNP significance compared to traditional studies that rely on single metrics. Furthermore, we employed advanced machine learning techniques alongside traditional methods, broadening the range of analytical tools available for SNP ranking.

A novel aspect of our study is the application of Pareto optimality in GWAS, which provides a multi-objective optimization framework that has not been extensively explored.

All these methods were applied to individual genotype and phenotype data downloaded from the Database of Genotypes and Phenotypes (*Mailman et al., 2007*). The biological interpretation of susceptibility loci thus identified as well as their therapeutic potentials were also provided and discussed.

## MATERIALS AND METHODS

### Data description and preparation

The individual genotype-phenotype data analyzed in the study were downloaded from the Database of Genotypes and Phenotypes (https://www.ncbi.nlm.nih.gov/gap). They are available under the accession number phs000615.v1.p1 once a data access request is approved through an authorized repository access system. The samples were collected by 13 genetic centers in the US and 11 ones in Europe and represented 6,505 patients with IS and 4,579 individuals without the disease (*Meschia et al., 2013*). From them, individuals aged over 55 years who identified themselves as "white" were selected. The processing of these data to meet the standard requirements of GWAS was described by us previously (*Khvorykh et al., 2023*). The final set of markers used in this research consisted of 883,749 SNPs and 159 indels. The number of individuals tested equaled to 5,581 (652 controls and 4,929 cases).

Assuming the additive inheritance model, we coded the genotypic data as 0, 1, 2, where the numbers corresponded to the number of copies of the less frequent allele in the whole dataset, and considered the SNPs to be categorical variables. We applied ML methods to both the raw data and the data after one-hot encoding, a transformation that converts categorical variables into a binary vector representation with only a single bit enabled per scaled variable.

There are approaches considering the genotypes in accordance with other data, *e.g.*, sex, age, clinical data, risk factors *etc*. In this research, we focused on genetics of the disease, thus correlating the genotypic data with the phenotype only. To minimize the potential effect of population stratification, the samples tested were preprocessed with principal component analysis and admixture clustering (*Alexander, Novembre & Lange, 2009*) to remove individuals exhibiting divergent ancestry.

This study does not require research ethics committees approvals, since it involves secondary analysis of anonymized data.

## Formal problem statement

The objective of this study is to rank SNPs by their significance, with those having a stronger association with IS receiving a higher rank. We explore two methodologies to achieve this.

The first one, classic, directly ranks features (SNPs) using $p$-values derived from GWAS. In such setting, SNPs are tested one by one, independently.

The second approach is based on ML classifiers. For this, we state our problem as a binary classification one. Data is described as follows. The set of all objects (individuals) is $X = \{\mathbf{x_1}, \mathbf{x_2}, ...., \mathbf{x_n}\}$. Set of all labels is $Y = \{y_1, y_2, ..., y_n\}$, where label $y_i \in \{0, 1\}$ equals to 1 if individual $i$ has experienced IS and 0 otherwise. The features are given by $f_j \colon X \to D_j, j = 1, \ldots, m$ where $D_j = \{0, 1, 2\}$ for each $j$. Values 0, 1 and 2 for $j$-th feature are obtained by encoding $j$-th SNP by comparing it to its reference value. This setup is well-suited for various classification models. We can select and train a model on this data, with the requirement that the chosen model must be capable of assigning importance scores (*Molnar, 2022*) to each feature (SNP) after training. More formally, for the $j$-th feature and a trained model $a$ it is possible to get a feature importance score $\text{FI}_a(j)$ with higher values indicating higher importance to the model. This capability enables the ranking of SNPs. The effectiveness of the classifier serves as an indicator of the quality of this ranking.

## Design of experiment

The main steps of data processing are given in Fig. 1. The genotype-phenotype data was initially analyzed using the chi-square test to rank the SNPs with $p$-values. The ranked list of SNPs was then applied to a greedy feature selection process under three machine learning approaches, each resulting in a list of important SNPs once the model was fitted. Furthermore, the Pareto optimality concept was applied to four measures of SNP importance, resulting in five lists of SNPs that were combined.

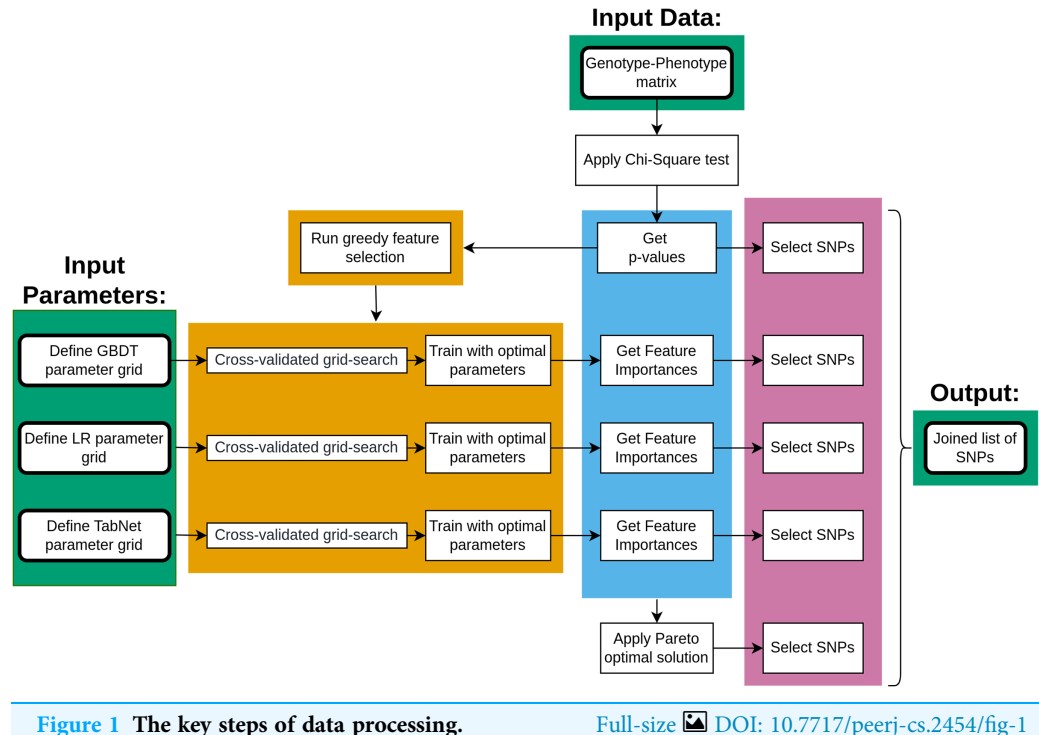

**Figure 1  The key steps of data processing.**

## Traditional GWAS approach

A widely used (*Clarke et al., 2011*) non-parametric approach for identifying SNPs associated with certain diseases in GWAS studies is the chi-squared test of independence. Every SNP is tested separately. The testing is done by computing real and expected allele frequencies between case and control groups. The chi-squared statistic and its associated *p*-value are then calculated. A lower *p*-value (or higher $-\log_{10}p$) indicates stronger evidence against the null hypothesis (which posits independence between the SNP and the disease), thereby signifying greater SNP importance and resulting in a higher rank. The results of application of chi-square testing followed by the correction of *p*-values for multiple comparisons with Bonferroni algorithm were described previously (*Khvorykh et al., 2023*). Here, we applied unadjusted *p*-values for SNP ranking. The results section presents the top-ranked SNPs and the Manhattan plot for the experiment, along with a reference to the code implementations.

## Machine learning approach

In this section we briefly describe the models we use in this study. They include the following: logistic regression, gradient boosting on decision trees (GBDTs), and TabNet.

A key reason for selecting these models is their ability to assign importance values to individual features, which in our case are SNPs. This feature allows us to gain insights into the relative significance of different genetic variants in the context of our study.

### Logistic regression

Logistic regression serves as a well-established "baseline" model for various applications in this domain (*Deloukas et al., 2013*; *Schunkert et al., 2011*).

It's a parametric ML model designed for solving classification problems. For binary classification setup we have $a_\theta : X \to [0, 1]$ where the output of the model $a_\theta$ on the object $\mathbf{x}$ is interpreted as probability of $\mathbf{x}$ belonging to positive class. The mapping to probabilities is done by a sigmoid function $a_\theta(\mathbf{x}) = (1 + \exp(-\theta^T \mathbf{x}))^{-1}$. Training is performed by minimizing logloss averaged among all objects $\frac{1}{N} \sum_{i=1}^{N} L(\mathbf{x_i}, y_i; \theta) = \frac{1}{N} \sum_{i=1}^{N} - y_i \log a_\theta(\mathbf{x}) - (1 - y_i) \log(1 - a_\theta(\mathbf{x})) \to \min$. The magnitude of each individual weight $\theta_j$ corresponding to a $j$-th feature of the object $\mathbf{x}$, $f_j(\mathbf{x})$, is interpreted as its importance and used further for feature ranking purposes. In other words $\mathrm{FI}_{a_\theta}(j) = |\theta_j|$.

### Gradient boosting on decision trees

Gradient boosting on decision trees (GBDT) is a widely used model for solving classification and regression problems. Its widespread is mostly due to its successful performances on various problems, mainly with tabular data (*Grinsztajn, Oyallon & Varoquaux, 2022*). To this day, GBDT remains a go-to approach for tabular datasets.

Two main ideas used in this approach are decision trees (DT) and gradient boosting technique (*Friedman, 2001*). Gradient boosting represents an ensemble of base models from some model family. Ensemble is done by getting the weighted average over all the base models. The idea is that at each next iteration, a new base model is trained on the residuals of the current ensemble. More formally, we initialize a base model $a_0(\mathbf{x})$ and ensemble $F_0(\mathbf{x})$ and start the iterative process, where $a_n(\mathbf{x})$ is trained on the residuals $r_{in} = \left[ \frac{\partial L(y_i, F(\mathbf{x_i}))}{\partial F(\mathbf{x_i})} \right]_{F(\mathbf{x}) = F_{n-1}(\mathbf{x})}$ and $F_n(\mathbf{x}) = F_{n-1}(\mathbf{x}) + c_n a_n(\mathbf{x})$. The computation of $c_n$ varies depending on implementation, the same goes for the stopping rule. The model family from which these base models are chosen are DTs. Without going into details, DTs are based on the successive application of the decision rules, which are usually in the form of the predicates; for example, $[f_j(\mathbf{x}) \leq t]$ for $j$-th feature and threshold $t$[1].

To measure the quality of the data split produced by predicates, the notion of Information Gain (IG) is often applied. First, the entropy of a prior (before applying the predicate) state is computed $H(m) = -\sum_{k=1}^{K} p_{mk} \log_2 p_{mk}$, where $K$ is the number of classes, $m$ represents the current node in a tree and $p_{mk}$ is a fraction of objects $\mathbf{x}$ in that node that have a class label $k$. Then, the entropy for a state after applying the predicate $H(m_L) = -\sum_{k=1}^{K} p_{m_L k} \log_2 p_{m_L k}$ and $H(m_R) = -\sum_{k=1}^{K} p_{m_R k} \log_2 p_{m_R k}$ is computed. Here, $m_L$ and $m_R$ represent the left and right children of $m$ with the right child satisfying the predicate at $m$ and the left one not. Finally, $IG(m) = H(m) - \frac{|X_{m_L}|}{|X_m|} H(m_L) - \frac{|X_{m_R}|}{|X_m|} H(m_R)$.

Among notable implementations of GBDT there are XGBoost (*Chen & Guestrin, 2016*), CatBoost (*Prokhorenkova et al., 2018*) and LightGBM (*Ke et al., 2017*). Each offers several ways to measure feature importance. Below we briefly describe two approaches used in this article.

---

[1] Expression in the form of Iverson notation for predicates $[f_j(x) \leq t]$ is evaluated either as 0 or 1, depending whether the condition inside brackets is false or true, respectively.

1) In CatBoost PredictionValueChange (PVC) measures for each feature how much the prediction changes on average when the feature value changes:

$PVC(i) = \sum_{trees, leafs_i} (v_1 - avr)^2 \cdot c_1 + (v_2 - avr)^2 \cdot c_2$, where $avr = \frac{v_1 \cdot c_1 + v_2 \cdot c_2}{c_1 + c_2}$, $i$ is a given feature; $c_1$, $c_2$ represent the total weight of objects in the left and right leaves, respectively; $v_1$, $v_2$ represent the formula value in the left and right leaves, respectively (*Dorogush, Ershov & Gulin, 2017*).

2) Average gain. For $j$-th feature it calculates the average IG among all predicates $[f_j(\mathbf{x}) \le t]$ in the ensemble where the feature is used. This means, given a trained model $a$, $FI_a(j) = \frac{1}{E_j} \sum_{t=1}^{T} \sum_{m=1}^{t_m} IG(m)[j \in index(m)]$ where $T$ is the number of trees in the ensemble, $t_m$ number of nodes in the tree $t$ and $E_j$ is the number of times $j$-th feature is used for the predicates in the ensemble.

### TabNet

TabNet is an interpretable canonical deep learning architecture designed specifically to address the challenges that deep neural networks (DNNs) encounter with tabular data (*Arik & Pfister, 2021*). According to the authors, the model was able to outperform leading GBDT approaches across a variety of benchmarks. At each decision step $t$, TabNet generates a feature selection mask $M_t$ using a sparsemax transformation of the raw attention scores $A_t$. The sparsemax function creates a sparse probability distribution, ensuring that only a subset of features is selected:

$$M_t = \text{sparsemax}(A_t)$$

The selected features $X \odot M_t$ are then used to contribute to the decision at step $t$. The cumulative importance score for each SNP $j$ is computed by aggregating these masks over all decision steps:

$$FI_{TabNet}(j) = \sum_{t=1}^{T} M_{t,j}$$

This score quantifies the frequency and significance of each SNP's selection throughout the model's decision process. We rank SNPs based on these cumulative importance scores to identify the genetic markers most influential in predicting the phenotype.

## Post-processing of the results of associative studies

### Pareto optimal solution

The concept of Pareto optimality comes from the field of multiobjective optimization. Let it be two or more objective functions: $f_1(\mathbf{x}), f_2(\mathbf{x}), \ldots, f_k(\mathbf{x})$, where $f_i : \mathbb{R}^n \to \mathbb{R}$ and $k \ge 2$. The task is to minimize all objective functions simultaneously. To exclude the trivial solution when every objective function attains its optimum, we assume objective functions to be at least partly conflicting. In this case, it is not possible to find a single solution that would be optimal for all objective functions simultaneously.

Pareto optimal solution is defined as a set of 'non-inferior' solutions in the objective space, when none of the objective functions can be improved without sacrificing at least of the other objective functions (*Miettinen, 1998*). In the other words, a decision vector $\mathbf{x}^* \in S$ is Pareto optimal if there does not exist another decision vector $\mathbf{x} \in S$ such that $f_i(\mathbf{x}) \leq f_i(\mathbf{x}^*)$ for all $i = 1, \ldots, k$ and $f_j(\mathbf{x}) < f_j(\mathbf{x}^*)$ for at least on index $j$.

The Pareto optimal solution was applied in this research to the results of four associative studies. More formally, for each feature indexed $j$ from 1 to $m$, we have $\mathrm{FI}_{a_1}(j)$, $\mathrm{FI}_{a_2}(j)$, $\mathrm{FI}_{a_3}(j)$, and $\mathrm{FI}_{a_4}(j)$, representing feature importances for chi-square testing, logistic regression, TabNet, and GBDT, respectively. This creates a 4-dimensional space of features from which the Pareto optimal ones were extracted for further analysis.

### Functional annotation

The polymorphisms selected as being associated with IS were annotated in terms of sequence ontology (SO) (*Eilbeck et al., 2005*) and genes with the snpEff 5.1 program (*Cingolani et al., 2012*), using the enclosed database GRCh37.87. The lists of genes thus obtained were tested for over-representation in the sets of human genes available in the Molecular Signature Database (MSigDB) (*Subramanian et al., 2005*). For the purpose, the hypergeometric test was applied. The False Discovery Rate (FDR) for significantly overrepresented genes was 0.05. For comparative analysis we also downloaded 1,159 genes associated with IS from DisGeNET (disgenet.org, accessed on 7 October 2022) (*Piñero et al., 2019*) and 323 genes from IDG program (druggablegenome.net, accessed on 21 March 2023).

## RESULTS

The source code of all the experiments, including the code to obtain the figures, hyperparameters selection and requirements are present at the GitHub repository: https://github.com/Stefan144/GWAS-ML. Python 3.10.12 (*Van Rossum & Drake, 2009*) was used alongside with R 4.2.1 (*R Core Team, 2022*) and Perl v5.34.0 (*Wall, Christiansen & Orwant, 2000*). The UpSet plot was created with UpSetR 1.4.0 package (*Gehlenborg, 2019*).

To address high dimensionality, we perform a feature selection procedure to extract the SNPs that influence the most on the quality of the logistic regression model. All SNPs were sorted by decreasing *p*-values obtained from the chi-square test described in "Traditional GWAS Approach". The top 1,000 SNPs were selected and a logistic regression model was fitted. Then the next 1,000 SNPs were added and the model retrained. This process was repeated until the model performance metrics stabilized. The model was trained with 5-fold cross-validation and calculation of binary classification metrics (precision, recall, F1). Cross-validation was employed to ensure the model robustness and generalizability. Given that the control group was the underrepresented class (11.67%) and metrics for cases were almost always good, we focused primarily on the metrics for controls to ensure balanced performance. The changes in F1 score for controls are illustrated in Fig. 2. As seen, it gradually increased with the increase in the number of SNPs reaching near permanent average values upon 15,000 SNPs. Since there was a subset of samples that demonstrated

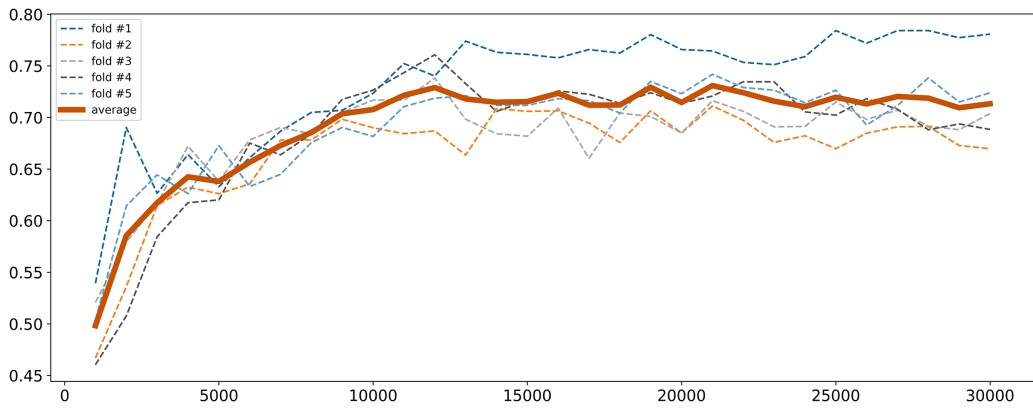

**Figure 2** **F1 score dynamic of logistic regression for class 0 depending on the number of SNPs selected.**

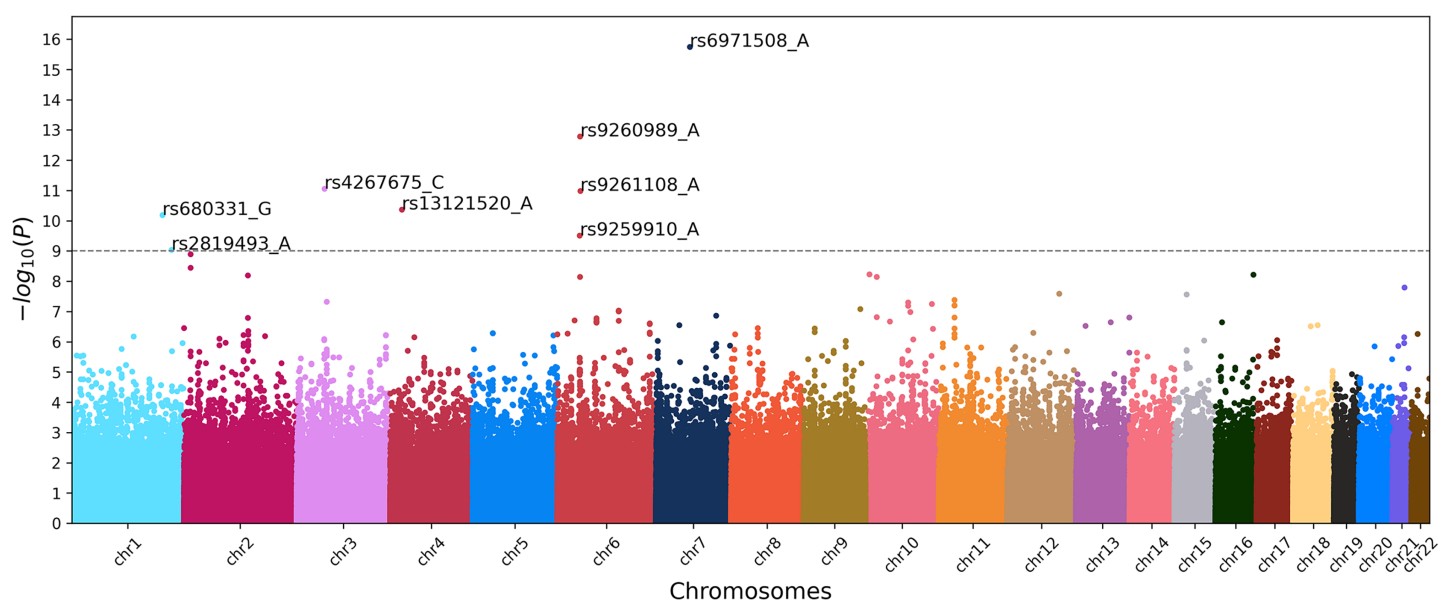

**Figure 3** **Manhattan plot of GWAS for ischemic stroke.** X-axis shows chromosomal positions. Y-axis shows $-log_{10}(p\text{-}values)$. The dashed line indicates the threshold $p$-value of $1.0 \times 10^{-9}$.

further increase in F1 score, finally we concluded that 30,000 SNPs were sufficient to achieve optimal model parameters.

## Loci associated to ischemic stroke

To identify the genetic loci associated with IS, we implemented three approaches based on ML and the classical GWAS based on chi-square test. The latter resulted in the Manhattan plot shown in Fig. 3.

To obtain results from the ML approaches, we first fine-tune the models using grid search combined with stratified cross-validation to determine the optimal set of hyperparameters. Cross-validation ensures that the model performs well on different

**Table 1 Metrics scores for models trained on optimal parameters.** Each score is an average across five folds accompanied by the standard deviation, shown after the ± symbol.

| | | Controls | | | Cases | | |
|---|---|---|---|---|---|---|---|
| Model | Accuracy | Precision | Recall | F1 score | Precision | Recall | F1 score |
| LR | 0.93 ± 0.07 | 0.74 ± 0.33 | 0.63 ± 0.33 | 0.68 ± 0.34 | 0.95 ± 0.04 | 0.97 ± 0.04 | 0.96 ± 0.04 |
| GBDT | 0.6 ± 0.06 | 0.12 ± 0.05 | 0.41 ± 0.2 | 0.19 ± 0.09 | 0.89 ± 0.03 | 0.62 ± 0.06 | 0.73 ± 0.05 |
| TabNet | 0.91 ± 0.04 | 0.72 ± 0.34 | 0.33 ± 0.25 | 0.44 ± 0.3 | 0.92 ± 0.03 | 0.99 ± 0.02 | 0.95 ± 0.02 |

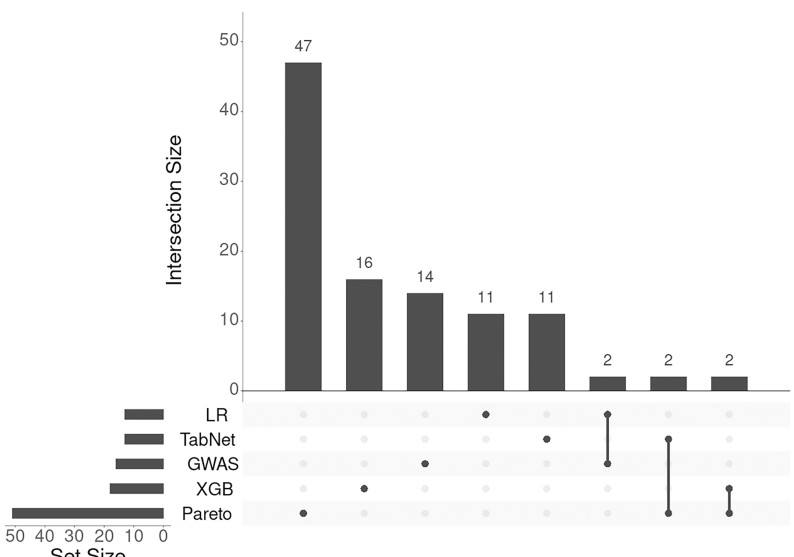

**Figure 4 UpSet plot of genes associated with IS obtained with five different approaches.**

subsets of the data, enhancing its generalizability and robustness. We use five-fold cross-validation. The optimal set of hyperparameters was determined based on the best average scores for the control group across all folds. This approach was necessary because the case group is overrepresented and consistently exhibits high performance metrics. The cross-validation performances for each model, trained with their optimal hyperparameters, are shown in Table 1. Each metric is averaged across five folds and accompanied by its standard deviation. Once these optimal hyperparameters are identified, we retrain the models from scratch on the whole dataset and extract the important features using methods specific to each model, as described in "Machine Learning Approach".

To integrate the results from the four methods (logistic regression, TabNet, gradient boosting, and GWAS $p$-values), we applied a Pareto optimal solution across these four dimensions of SNP importance. This approach served as a consensus among the methods, leading to the identification of 131 genes and loci (corresponding to 105 Entrez gene names) associated with an increased risk of IS onset. The results of annotation of SNPs and the list of gene symbols are given in the Supplemental Files selected-snps-annotation.xlsx and genes-01.xlsx, respectively. As seen from Fig. 4 the lists of associated genes being almost unique for each approach considered.

The SO terms assigned by SnpEff program for all approaches presented were mainly as follows: intergenic_region, intron_variant, and intragenic_variant. We annotated all 883,749 SNPs available under this research and found these terms to account for 90% of SO terms of all SNPs. Therefore, the distribution of SO terms of selected SNPs followed the general distribution of these terms.

There was one exception among SO terms occurred in the results of annotation revealed by TabNet. The SNP rs1702003 ($p$-value of chi-square test = $7.9 \times 10^{-4}$) was annotated as missense variant in *ACOT11* gene. This gene encodes the member of the family of acyl-CoA thioesterase enzymes that play an important role in the metabolism of fatty acids by catalyzing the conversion of activated fatty acids to the corresponding non-esterified fatty acid and coenzyme A (*Kirkby et al., 2010*).

The gene over-representation analysis did not reveal any known sets of genes from MSigDB, in which the genes identified by us as associated with IS were over-represented significantly (FDR < 0.05).

The intersection of candidate genes with a list of genes downloaded from DisGeNET revealed three ones that had been described as associated with IS: *UBQLN1* (rs10746718, intron_variant) in the case of Pareto optimal solution, as well as *TRPS1* (rs2575911, intergenic_region) and *MUSK* (rs474186, intergenic_region) for logistic regression. *UBQLN1* encodes ubiquilin-1 that is responsible for the degradation of misfolded proteins (*Ko et al., 2004*). The SNP rs2575911 lies upstream to the gene *TRPS1*. According to http://grch37.ensembl.org, this SNP is inside of open chromatin region (ENSR00002075367) and its allelic variants change slightly the expression of this gene in whole blood ($p$-value = 0.018, effect size = 0.052). The gene *TRPS1* encodes the zinc finger transcription factor initially associated with the regulation of chondrocytes and the development of perichondrium (*Napierala et al., 2008*). The SNP rs474186 lies downstream of the MUSK gene that encodes a muscle-specific tyrosine kinase receptor playing a crucial role in the formation of neuromuscular junctions, including the clustering of acetylcholine receptors. Mutations in *MUSK* have been associated with congenital myasthenic syndromes (*Herbst, 2020*).

## Druggable genome genes

The intersection of IDG genes with the genes obtained by annotation of all SNPs available in our research revealed 278 common genes and 12,726 SNPs annotated to them (Supplemental File). The intersection of genes identified in this research as being associated with the onset of IS with IDG genes revealed one common gene *GPR26*. It was found by chi-square test (rs34201757, $p$-value = $7.4 \times 10^{-08}$). The gene encodes a G-protein coupled receptor (GPCR) that belongs to a large family of integral membrane proteins. GPCRs are involved into conversion of different extracellular stimuli (neurotransmitters, hormones, proteins, peptides, small-molecule odorants, pheromones, light *etc.*) to intracellular responses.

It is interesting to mention here that there are only two IDG genes–*TAOK1* and *GPR65* (synonym *TDAG8*) among 1,195 IS genes presented in disgenet.org project. Both have to do with the IS induced in rats. TAOK1 ameliorated IS was related to cell injury by

decreasing the pro-inflammatory factors and apoptosis in them (*Li et al., 2019*). Cerebral infarction and dysfunctional behavior were exacerbated with the deficiency of TDAG8. The underlying mechanism involves supposedly the inhibition of some microglial functions by TDAG8 under an acidic environment developed after IS (*Sato et al., 2020*).

## DISCUSSION

The idea behind this research was to develop bioinformatics methods alternative to classical GWAS. Historically, the first GWAS was based on statistical testing of individual SNPs. Alternatively, the whole set of SNPs can be examined together. This can be achieved by training a predictive model with supervised ML algorithms. Then, the association of SNPs to disease is estimated as information contribution to the target variable within the predictive model trained. An alternative way to feature interpretability of machine learning models could be a model-agnostic approach like Shapley values. However, their full evaluation is exhaustive, which is a serious limitation in the case of hundreds of thousands of SNPs even for simple rule-based schemes (*Ignatov & Kwuida, 2022*).

Following this concept, we implemented three approaches based on ML methods: logistic regression, GBDTs, and the interpretable neural network TabNet. We also applied chi-square testing. Finally, the Pareto optimal solution was implemented to the results of ML and classical GWAS. In total, five approaches resulted in 131 genes and loci associated with IS. The lists of genes obtained were almost unique. This may be due to the association of SNPs with the disease being measured differently by each approach. Two aspects of differences can be considered.

Firstly, how the association of individual SNP was estimated. In general, both classical and ML-based GWAS apply the same concept: estimate the association of SNP with disease and select the most reliable ones. However, the measures of association differ: $p$-value of chi-square test, coefficients of logistic regression, feature importances of XGBoost and TabNet. All these measures are model-dependent. They are specific to a particular algorithm. It implies a reasonable question, which measure is better? However, this task was out of the scope of this research. It can be solved with the use of data having known association with phenotype. Meanwhile, we observe the usefulness of these measures in identifying the genes associated with IS.

Secondly, how the SNPs relate to each other. In the case of chi-square test, each SNP was processed separately, the $p$-value represented the strength of correlation of individual SNP with outcome. In the case of ML methods, all SNPs were processed simultaneously, the scores reflected the contribution of each SNP to the outcome as a part of the whole set of SNPs in the model. Should the results obtained by two methodologies be the same is a rather complicated question. SNPs might also be correlated and uncorrelated because of LD. How does this phenomenon influence on the results of ML methods requires additional investigation.

Taking into account two aspects described above, we hypothesized the five approaches applied to be like planes intersecting multidimensional space of genotypes of sick and healthy individuals. Each of the technique considered resulted in a set of significant SNPs

bringing certain information about underlying biological processes. Therefore, we analyzed all sets of genes thus produced. Their comparison with those described in the literature revealed the four ones with good level of evidence of possible participation in pathogenesis of IS.

The first interesting observation was brought by TabNet. Among Top-10 SNPs contributed the most to the outcome variable there was rs1702003 identified as missense variant in *ACOT11* gene. This gene participates in metabolism of fatty acids and thus potentially can influence on the onset of IS. *ACOT11* was found to regulate non-shivering thermogenesis by limiting the oxidation of fatty acids derived from the lipolysis of lipid droplets in mouse brown adipose tissue (*Okada et al., 2016*). The deletion of this gene in mice increased the energy expenditure and protected animals against diet-induced obesity and insulin resistance. The data suggested that ACOT11 decreases energy consumption and conserves calories. It is important to note that *ACOT11* is induced not only in response to cold exposure but also with a high-fat diet too. In the setting of nutritional excess, caloric conservation can lead to development of obesity. Moreover, free fatty acids are pro-inflammatory molecules and their overproduction by ACOT11 may provoke inflammation, insulin resistance and endoplasmic reticulum stress, which resulted in further disturbance of lipid and glucose metabolism. Obesity, blood lipids, type 2 diabetes and atherosclerosis are the risk factors of IS and the traits that can share genetic variation with it (*Malik et al., 2018*). We believe these observations support associations found between IS and *ACOT11*.

The second promising finding was made with Pareto optimal solution. Such an approach was already applied to SNP datasets in population and medical genetics. Pareto optimality allowed to subset SNPs that discriminated the ethnic group better than without it. The authors applied three criteria model MI-PC1-PC2, where MI is mutual information, while PC1 and PC2 are principal components of principal component analysis (*Gumus, Gormez & Kursun, 2013*). The same problem was tackled with three different aggregation methods, including Pareto optimal solution. In this case, each SNP was evaluated by four criteria. Pareto optimality yielded the best classification accuracy (*Gormez et al., 2013*). Finally, using two-objective functions and ant colony optimization algorithm, other authors detected the SNP epistasis in simulated and real Alzheimer's disease GWAS data. The first objective function was Akaike Information Criterion applied for logistic regression, the second objective function used frequency measurements from mutual information theory (*Yuan, Yuan & Huang, 2017*). These three examples showed that Pareto optimality can be applied to SNPs characterized by different criteria to subset polymorphisms with optimal characteristics.

So far we did not notice the application of Pareto optimality concept to study the genetics of IS in the literature. We characterized each SNP by associations with IS obtained with four methods and applied Pareto optimal solution. This resulted in SNP rs10746718 that was in the list of TabNet and XGBoost methods, but was not among the top loci. SNP rs10746718 is located in the intron of the *UBQLN1* gene dealt with the protein degradation. Genetically modified mice with increased expression of *UBQLN1*

demonstrated reduced volume of the infarct and recovered motor function more rapidly after IS induced by middle cerebral artery occlusion than wild-type littermates, while opposite effects were observed in animals with decreased expression of this gene. Moreover, knock-out of this gene resulted in a significant accumulation of ubiquitinated proteins in the mouse brain. The authors proposed that ubiquilin-1 protects the mice from neuronal injury caused by IS *via* facilitating the removal of damaged proteins (*Liu et al., 2014*). This way the ubiquilin-1 was also connected with Alzheimer and other neurodegenerative diseases as well as cancer (*Li et al., 2019*). However, how could it affect the incidence of IS and explain the association found? One can propose that it is the consequence of its properties to protect cells from oxidative stress that was demonstrated in experiments with menadione–an intracellular oxyradical producing agent, whose effects were less pronounced in cells with increased expression of *UBQLN1* (*Liu et al., 2014*). Emerging evidence indicates that oxidative stress promotes many pathological conditions. For example, elevated levels of reactive oxygen/nitrogen species can disrupt endothelial functional ability, induce atherosclerosis, and impair vasculature all of them are common characteristics of cerebrovascular diseases, including IS (*Kumar et al., 2023*). The protective effect of UBQLN1 in oxidative stress may be associated with the enhanced removal and degradation of potentially toxic damaged proteins by the proteasome or autophagosome machinery (*Liu et al., 2014*; *Lin et al., 2021*). The other option seems to be attributed to the role of UBQLN1 in the removal of misfolded proteins from the pool of the proteins prepared in the endoplasmic reticulum for trafficking to other cellular compartments, or for secretion across the plasma membrane, thereby maintaining normal metabolism both of dietary components that may enhance both the production of reactive oxygen species (ROS), *e.g.*, elevated blood lipids, and ROS themselves (*Mohanty et al., 2002*; *Yang et al., 2008*; *Hetz, Zhang & Kaufman, 2020*). The same way is used to control functioning of the electron transport chain of mitochondria, one of the main sources of ROS in the cells (*Itakura et al., 2016*).

The third observation worth to be discussed is the association of rs2575911 with IS identified with logistic regression. According to the Ensembl genome browser (*Cunningham et al., 2021*), the variant lies upstream to the gene *TRPS1* and within a regulatory element found computationally. Being a transcription factor, TRPS1 was initially associated with the regulation of chondrocytes and the development of perichondrium. The loss of its function has been linked with the development of tricho-rhino-phalangeal syndrome (TRPS), a rare autosomal dominant disorder characterized by several skeletal and facial abnormalities and sparse scalp hair (*Napierala et al., 2008*). However, recent data suggested a multifaceted functionality for TRPS1. It plays the role in the development of multiple tissues, metabolic disorders as well as malignant tumors (*Yang et al., 2022*). Two polymorphisms within this gene were found to be associated with the risk of development of IS (rs2954029) and coronary heart disease (rs2954029 and rs231150) in Chinese population (*Zhang et al., 2019*). As SNP rs2954029 was also correlated with serum lipid levels, increased risk of the diseases might be due to dyslipidemia and accompanying atherosclerosis, which is the pathological basis of both

coronary heart disease and IS. We linked *TRPS1* with the onset of IS in European population for the first time. However, the underlying mechanism is to be clarified.

Another gene associated with IS from logistic regression and presented in DisGeNET, *MUSK*, seems to be included in this database by a mistake. Previously it had the name *FADS1*, but today *FADS1* denotes a different gene that is associated with IS due to its role in lipid metabolism (*Yuan et al., 2019*). As the SNP rs2575911 was annotated as intergenic SNP, its association with IS seemed to be attributed to the second element of the pair, *LPAR1* gene, which was not presented in DisGeNET. This gene encodes a lysophosphatidic acid (LPA) receptor belonging to a group of receptors from the G protein-coupled receptor superfamily (*Xiang et al., 2020*). It was shown that LPAR1 plays a critical role in microglial activation and brain damage after transient focal cerebral ischemia in mice (*Gaire et al., 2019*). It was also associated with hypertension and risk of thrombosis (*Xu et al., 2015*; *Xiang et al., 2020*). The latter occurs due to platelet activation induced by LPA, a molecule formed during mild oxidation of low density lipoproteins (*Rother et al., 2003*).

We also would like to emphasize the gene *GPR26* from the IDG project. The association of this gene with IS has been shown by us on the same dataset previously (*Khvorykh et al., 2023*). The gene belongs to a large family of GPCRs. Several such genes were found to be associated with nervous system disorders and onset of IS as well as its functional outcome. GPCR genes are also likely to be involved in the pathophysiology of IS through their influence on risk factors such as hypertension, atherosclerosis, and heart failure (*Vahidinia et al., 2021*; *Kaur et al., 2023*). This role seems to be attributed to *GPR26*, whose deficiency has been associated with obesity and diabetes, both of which are known risk factors for IS (*Chen et al., 2012*; *Kichi et al., 2022*). In patients with type 2 diabetes, *GPR26* is down-regulated due to chronic hyperglycemic conditions and accompanied by increased production of reactive oxygen species, pro-inflammatory monocyte activation and adhesion to endothelial cells (*Kichi et al., 2022*). The underlying mechanism of *GPR26* association with ischemic stroke is to be clarified. Nevertheless, the described relationships suggest common genetic determinants by which different disorders can influence on the pathogenesis of IS as well as the potentials of *GPR26* as the therapeutic target in both IS and its risk pathological conditions.

Upon comparing the genes associated with IS identified in our study with existing literature, several key insights emerge. Primarily, the associations with IS predominantly occur *via* well-established risk factors such as obesity, dyslipidemia, type 2 diabetes, and hypertension, among others. Modifiable factors are known to contribute to approximately 90% of the stroke risk (*O'Donnell et al., 2016*). Considering stroke heritability, estimated at around 40%, suggests shared genetic influences with some of these risk factors (*Malik et al., 2018*). Understanding variations in these genes holds potential for predicting stroke risk through the development of polygenic risk scores and identifying new targets for stroke prevention and treatment (*Abraham, Rutten-Jacobs & Inouye, 2021*; *Dichgans et al., 2021*).

Interestingly, one of the genes identified in our study, *GPR26*, is already recognized as a potential therapeutic target. We posit that other genes may also represent promising sites for therapeutic intervention, potentially through their influence on corresponding micro

or circular RNAs (*Rupaimoole & Slack, 2017*; *Liu et al., 2022*). This insight underscores the potential for leveraging genetic variations in the development of targeted stroke therapies.

This research has some limitations. Firstly, we analyzed the genotypes of individuals of European ancestry. The GWAS risk variants for IS indicate a distinct population stratification (*Sukumaran, Nair & Banerjee, 2024*). Secondly, only an additive model of inheritance of the trait under investigation was tested by us. This model is widely used and even considered the most preferred for obtaining the best performances across multiple algorithms and data sets (*Mittag, Römer & Zell, 2015*). However, since the inheritance of multifactorial diseases is uncertain, other models of inheritance (dominant, recessive, and overdominant) may be useful, but they may result in different associations. Thirdly, the SNPs influence gene functions differently. The weaker the effect of SNPs, the larger the cohort of individuals should be to catch such SNPs. Taking into account the aspects just mentioned, we believe our research encourages ML-based GWAS on independent cohorts, where people of other ancestries will be studied, other inheritance models tested, and larger groups of individuals involved. The subsequent research will also be required for functional testing of candidate genes in an appropriate model organism or *in vitro*. To conclude, the results presented in this study deserve the attention of researchers interested in the elaboration of new approaches to discover the genetic loci associated with the multifactorial diseases.

## CONCLUSIONS

In this study, we explored a novel methodology for GWAS utilizing machine learning techniques. Our approach involved constructing predictive models using SNPs as categorical features, followed by ranking them based on their contribution to the target variable and selecting the most informative ones. We assessed the association of SNPs with the trait using various metrics, including the $p$-value from the chi-square test, the absolute values of weights from logistic regression, the attention scores from the interpretable neural network TabNet, and the cumulative information gain from GBDTs.

Viewing each of these metrics as an objective in a multi-objective optimization problem, we proposed utilizing the Pareto-optimal solution to select significant loci. When applied to genotypic data from 5,581 individuals with and without IS, our methods collectively identified 131 genes and loci potentially associated with IS onset. The lists of genes were almost unique for each of the methods used. Nevertheless, a subsequent literature review of several of these genes confirmed their potential involvement in stroke development.

To further elucidate the effects of these genes, a systematic investigation of their association with IS-related traits is warranted. Additionally, one gene identified in our study, *GPR26*, was among the IDG genes, suggesting its product as a potential drug target. However, it is plausible that other discovered genes may also be considered as potential drug targets in the future. This could enhance our ability to control IS risk through more personalized correction of stroke risk factors.

We believe that the gene search methodology proposed and investigated in this study holds promise for further development and application to GWAS data of other diseases.

### Funding

The study was funded by the Russian Science Foundation, grant number 23-14-00131 (training, evaluation and implementation of machine learning models, genome-wide association study, functional interpretation of the results obtained). The work was also an output of a research project implemented as part of the Basic Research Program at the National Research University Higher School of Economics (design of experiments and selection of methods). The funders had no role in study design, data collection and analysis, decision to publish, or preparation of the manuscript.

### Grant Disclosures

The following grant information was disclosed by the authors:
Russian Science Foundation: 23-14-00131.
Basic Research Program at the National Research University Higher School of Economics.

### Competing Interests

The authors declare that they have no competing interests.

### Author Contributions

- Stefan Nikolić conceived and designed the experiments, performed the experiments, analyzed the data, performed the computation work, prepared figures and/or tables, authored or reviewed drafts of the article, and approved the final draft.
- Dmitry I Ignatov conceived and designed the experiments, analyzed the data, prepared figures and/or tables, authored or reviewed drafts of the article, weekly mentoring sessions, and approved the final draft.
- Gennady V Khvorykh conceived and designed the experiments, performed the experiments, analyzed the data, performed the computation work, prepared figures and/or tables, authored or reviewed drafts of the article, and approved the final draft.
- Svetlana A Limborska conceived and designed the experiments, analyzed the data, authored or reviewed drafts of the article, conceptual planning and organising working meetings, and approved the final draft.
- Andrey V Khrunin conceived and designed the experiments, analyzed the data, authored or reviewed drafts of the article, data and finance acquisition, and approved the final draft.

### Data Availability

The individual genotype-phenotype data are available at the international database of genotypes and phenotypes dbGaP: phs000615.v1.p1. They were collected by 13 genetic centers in the US and 11 ones in Europe and represented about 6000 patients with IS and the same number of people without this disease.

https://www.ncbi.nlm.nih.gov/projects/gap/cgi-bin/study.cgi?study_id=phs000615.v1.p1

The code is available at GitHub and Zenodo:

- https://github.com/Stefan144/GWAS-ML

- Stefan144, & Gennady Khvorykh. (2024). inzilico/GWAS-ML: v1.0 (v1.0). Zenodo. https://doi.org/10.5281/zenodo.12800611.

The preprocessing data is available at GitHub: https://github.com/inzilico/GWAS-ML.

## Supplemental Information

Supplemental information for this article can be found online at http://dx.doi.org/10.7717/peerj-cs.2454#supplemental-information.

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
