# Peer review of "Genome-wide association studies of ischemic stroke based on interpretable machine learning"

_PeerJ Computer Science, doi:10.7717/peerj-cs.2454_

## Round 0.1 · original submission · Major Revisions

Dear Authors,

Thank you for your manuscript. While the reviewers found it interesting, two key revisions are necessary:

Detailed Methods: Provide a more thorough description of methods and models. This allows for replication and future development.

Public Code: Open-source the code. This adheres to open-science principles and enables others to verify and utilize your work.

Addressing these points will strengthen your paper for publication.

Sincerely,

Marta Lovino

Reviewer 1 ·

Basic reporting

All comments have been added in detail to the 4th section called additional comments.

Experimental design

All comments have been added in detail to the 4th section called additional comments.

Validity of the findings

All comments have been added in detail to the 4th section called additional comments.

Additional comments

Review Report for PeerJ Computer Science
(Genome-wide association studies of ischemic stroke based on interpretable machine learning)

1. Within the scope of the study, genome-wide association studies were carried out using various machine learning models such as decision tree and logistic regression, in addition to classical statistical tests.

2. The dataset used is open source, and in a previous study of the authors, this dataset was processed in accordance with genome-wide standards, and before being used in this study, it was again subjected to preprocessing processes, followed by three different machine learning approaches: Logistic Regression, Gradient Boosting on Decision Trees and TabNet. used. Although there are many different machine learning models in the literature that can be used to solve this problem, it should be clearly stated why these models are preferred and their originality.

3. Although the problem is discussed and the literature section is at a certain level in the introduction, a literature table with columns such as "dataset, problem, pros and cons, results, ..." should be added to make the comparison with the literature clearer, and also "materials and methods" section, it is suggested that the study's main contribution to the literature, its difference from the literature, and its originality should be expressed more clearly in items.

4. The problem addressed in the study and its realization within the scope of a research project are very valuable. When examined in this context, it is clear that it will make a significant contribution to the literature.

As a result, it is recommended to pay attention to the above-mentioned parts in order to further increase the post-publication citation potential of the study and to express its originality more clearly.

Reviewer 2 ·

Basic reporting

1. Please increase the font size of the figures. The texts are difficult to read without zooming in.
2. I can not find the data and the Python script that generates results. I suggest the author put them on GitHub or other websites.
3. Please provide the versions of the Python package and R package used in this work.

Experimental design

The author does not describe the splitting of the data set.
I cannot know how the author split the dataset into the training set, validation set, and testing set.
The author also does not introduce how they use five-fold cross-validation. I only find five folds in Figure 1. Most detail information is missing.

Validity of the findings

I cannot evaluate the validity of the findings because the author does not provide sufficient information of the machine learning model training and testing process. I suggest the author provide these details so that I can re-evaluate the validity of the result.

Reviewer 3 ·

Basic reporting

Strengths:
Clear and unambiguous, professional English: The language used throughout the paper is clear, concise, and appropriate for a scientific audience.
Literature references and field background: The introduction provides a sufficient background on GWAS, including its limitations and the potential of machine learning approaches. Relevant literature is cited to support the claims and provide context for the study.
Professional article structure: The paper follows a typical structure for scientific articles, including introduction, methods, results, discussion, and conclusion sections.
Figures and tables: The paper includes several figures and tables that effectively present the results and aid in understanding the methodology. The Manhattan plot and UpSet plot are particularly informative.

Weaknesses:
Raw data sharing: The paper does not mention whether the raw data is publicly available. Sharing raw data is crucial for reproducibility and further analysis by the scientific community.
Self-contained with relevant results to hypotheses: While the paper presents results relevant to the main hypothesis of identifying genetic loci associated with IS, some aspects could be improved. The discussion of the identified genes' functions and potential roles in IS could be more thorough.
Formal results and proofs: The paper does not present formal definitions or proofs, as it focuses on the application and interpretation of existing statistical and machine learning methods. However, it clearly explains the chosen methods and their rationale.

Suggestions for improvement:
Data availability: Clearly state whether the raw data is available and, if so, provide instructions for accessing it.
Deeper discussion: Expand the discussion on the biological functions of identified genes and their potential roles in IS pathogenesis. Explore potential connections between genes and established IS risk factors.
Methodological details: Provide more details on the hyperparameter optimization for the machine learning models.

Overall:
The paper demonstrates good basic reporting with clear language, relevant background information, and a professional structure. However, it would benefit from improved data sharing practices and a more in-depth discussion of the biological implications of the findings.

Experimental design

Strengths:
Original primary research within Aims and Scope: The study appears to be original primary research focusing on the genetic basis of ischemic stroke, which aligns with the scope of many journals in the fields of genetics, bioinformatics, and medical research.
Research question well defined, relevant & meaningful: The research question of identifying genetic loci associated with increased risk of IS is clearly defined, relevant to the field, and has the potential to advance our understanding of the disease and contribute to the development of preventive or therapeutic strategies.
Identified knowledge gap: The introduction clearly explains the knowledge gap in understanding the genetic basis of IS, despite the identification of some risk loci through traditional GWAS.
High technical standard: The study utilizes established statistical methods and machine learning algorithms. It also employs a consensus-building approach using the Pareto-optimal solution, enhancing the robustness of the findings.

Weaknesses:
Methods description: While the methods section provides a general overview of the techniques used, some aspects lack sufficient detail for complete replicability. For instance, the specific hyperparameters used for the machine learning models and the criteria for selecting the optimal set of SNPs are not fully described.
Ethical considerations: The paper does not explicitly discuss ethical considerations related to the use of human genetic data. Although the data was obtained from a public database, it is essential to acknowledge ethical aspects and ensure compliance with relevant regulations and guidelines.

Suggestions for improvement:
Detailed methods: Provide a more detailed description of the methodological steps, including specific parameters and criteria used for data processing, SNP selection, and model optimization. This would allow other researchers to replicate the study more accurately.
Ethical statement: Include a statement addressing the ethical considerations of using human genetic data, including informed consent, data anonymization, and compliance with relevant ethical guidelines.
Model interpretability: While the paper mentions the use of interpretable machine learning models, it could benefit from a more in-depth discussion of how the models were interpreted and how these interpretations contributed to the biological understanding of the results.

Overall:
The experimental design is sound and addresses a relevant research question in the field of IS genetics. However, improvements in methodological transparency and ethical considerations would further strengthen the study.

Validity of the findings

Strengths:
Replication encouragement: The paper acknowledges the need for further validation and encourages replication studies. It proposes investigating the identified genes' association with IS-related vascular traits, which provides a clear rationale and potential benefit to the literature.
Conclusions linked to research question: The conclusions directly address the original research question of identifying genetic loci associated with IS. They summarize the main findings and highlight the potential of the combined GWAS and machine learning approach.
Conclusions limited to supporting results: The paper avoids overstating the findings and acknowledges the limitations of the study. It appropriately frames the identified genes as "candidates" requiring further investigation and validation.

Weaknesses:
Data availability: While the paper mentions using data from dbGaP, it lacks specific details on data access and does not provide the processed data used in the analyses. This limits the ability of other researchers to directly validate the findings.
Statistical soundness and controls: The paper describes the use of statistical tests and machine learning methods but does not provide extensive details on the statistical analyses or control measures implemented. More information on these aspects would strengthen the confidence in the validity of the findings.

Suggestions for improvement:
Data and code sharing: Make the processed data and code used in the analyses publicly available through a suitable repository. This would enable independent verification and facilitate further research.
Detailed statistical reporting: Provide more comprehensive information about the statistical methods used, including specific tests, parameters, and measures of significance. Discuss potential confounding factors and how they were addressed.
Replication studies: Conduct or encourage independent replication studies using different populations or datasets to further validate the identified genetic associations.

Overall:
While the conclusions are well-stated and appropriately limited to the supporting results, the lack of readily accessible data and detailed statistical reporting makes it difficult to fully assess the validity of the findings. Improving data and code sharing practices, along with more comprehensive reporting of statistical analyses, would significantly enhance the study's transparency and reproducibility.

Additional comments

Comparison with existing literature: While the paper identifies some previously reported genes associated with IS, a more comprehensive comparison with existing literature on IS genetics could be beneficial. This would provide a broader context for the findings and help to assess their novelty and significance within the field.

Functional validation: The paper focuses primarily on identifying candidate genes, but it would be valuable to explore functional validation experiments. These could involve in vitro or in vivo studies to investigate the biological mechanisms by which the identified genes contribute to IS risk.

Clinical implications: While the paper touches on potential therapeutic implications, a more detailed discussion of how the findings could be translated into clinical practice would be valuable. This could include exploring the development of genetic risk prediction models or identifying potential drug targets for personalized medicine approaches.

Exploration of other machine learning methods: The paper investigates three different machine learning algorithms, but there are many other potential methods that could be explored. Investigating and comparing a wider range of algorithms could provide further insights and potentially identify additional genetic associations.

Integration of multi-omics data: Future studies could consider integrating genetic data with other omics data, such as transcriptomics or proteomics. This could provide a more comprehensive understanding of the molecular mechanisms underlying IS and identify additional biomarkers or therapeutic targets.

Overall:
This study presents a promising approach for investigating the genetics of IS and highlights the potential of combining GWAS with machine learning methods. By addressing the limitations and incorporating the suggestions mentioned above, future research in this area can build upon these findings and contribute significantly to our understanding and treatment of ischemic stroke.

---

## Round 0.2 · Minor Revisions

The revisions have substantially enhanced the quality of the manuscript. Both reviewers have provided valuable feedback, and I am pleased to see how well you have addressed their concerns. That said, there are still a few outstanding issues. I would strongly encourage you to carefully consider the comments from Reviewer 2 regarding code training and those from Reviewer 3 related to Basic Reporting. Addressing these points will further strengthen your manuscript.

Reviewer 1 ·

Basic reporting

All comments have been added in detail to the last section.

Experimental design

All comments have been added in detail to the last section.

Validity of the findings

All comments have been added in detail to the last section.

Additional comments

Review Report for PeerJ Computer Science
(Genome-wide association studies of ischemic stroke based on interpretable machine learning)

Thanks for the revision. As a result of the first referee review, both the relevant changes made to the paper and the responses to the referee comments are sufficient. For this reason, I recommend that the paper be accepted. I wish the authors great success in their future projects and papers. Kind regards.

Reviewer 2 ·

Basic reporting

Most issues have been improved. Please see "additional comments".

Experimental design

Most issues have been improved. Please see "additional comments".

Validity of the findings

Most issues have been improved. Please see "additional comments".

Additional comments

In the code file for training (e.g linear regression: lr.ipynb), I cannot find the tutorial to generate "stroke_data" for the training model. I suggest the author provide this result so the reader can reproduce the code.

Reviewer 3 ·

Basic reporting

The paper demonstrates generally sound basic reporting. The language is clear, the structure is logical, and relevant references and data accessibility enhance the scientific rigor. However, further elaborating on certain aspects, like the background information, methodological choices, and SO term analysis, could strengthen the clarity and comprehensiveness of the reporting.

1. The introduction could potentially benefit from a more detailed overview of the existing knowledge gap regarding the genetic bases of ischemic stroke (IS). Although it mentions 90 identified genetic loci, further elaboration on the limitations of previous studies could stregthen the rationale for this specific research.

2. While the key steps of data processing are presented in a figure, further detail regarding the rationale and application of each ML technique (Logistic Regression, GDBTs, TabNet) might be beneficial. Specifically, clarifying the advantages of these methods over more traditional GWAS approaches would add clarity.

3. The paper states that 30,000 SNPs were chosen baased on stabilized performance metrics. A more elaborate explanation of the selection criteria and a demonstration (perhaps graphically) of the metric stabilization could stregthen this selection.

4. While the paper notes the prevalence of common SO terms, it primarily focuses on a single exception (missense variant). A broader analysis of the distribution and potential significance of SO terms associated with the identified SNPs could enrich the results section.

Experimental design

The experimental design presents a well-considered approach to GWAS, effectively leveraging the strengths of machine learning models to uncover the genetic complexities of IS. The multi-model approach, feature selection strategy, and control for population stratification demonstrate scientific rigor. However, certain aspects could benefit from further clarification and expansion, particularly concerning the justification for model choices, comprehensive performance reporting, and direct comparisons with traditional methods.

1. While the chosen models (logistic regression, GDBTs, TabNet) are widely used, a more detailed rationale for their selection would be beneficial. Explaining how these models specifically address the challenges of GWAS data analysis (high dimensionality, potential for non-linear relationships, etc.) would solidify the methodological choices.

2. While the paper mentions a 5-fold cross-validation and focuses on control group metrics, a more detailed presentation of model performance metrics (AUC, precision, recall, specificity, etc.) for all models and both classes (cases and controls) would offer a more complete assessment of model effectiveness.

3. While stablizaition of performance metrics is a valid criterion, the selection of 30,000 SNPs might be further justified. Demonstrating how other thresholds impact model performance or demonstrating metric stablization more concretely (potentially with a figure) could strengthen the decision.

4. Including a comparative analysis of the genes identified using ML methods against those solely identified by chi-square testing would be insightful. It could reveal the added value and potential for novel discoveries using ML techniques for GWAS.

5. The research is focused on the onset of IS. Considering other relevant phenotypes, like stroke severity or subtypes, could broaden the impact and generalizability of the findings.

Validity of the findings

While the paper presents a novel methodology for GWAS analysis using machine learning and achieves intriguing results, certain aspects require careful consideration to assess the validity of the findings.

1. The study relies heavily on bioinformatic analysis and literature review. Further experimental validation is crucial to confirm the roles of these identified genes in IS. This could invovle: (1) model organisms: Studying the function of these genes in IS-relevant animal models (e.g., mice); (2) cellular studies: Investigating the impact of genetic variations in these genes on relevant cellular processes in vitro.

2. The findings are based on data from individuals of European ancestry. The generalizaibility to other populations warrants investigation due to known ethnic variations in genetic risk factors for IS. The paper highlights potential associations between genes and IS onset, but it does not conclusively establish causality. Further research is needed to explore functional mechanisms and confirm the directions of effects.

3. The emphasis on the ACOT11 missense variant identified by TabNet seems dispropoertionate, especially considering the dominance of non-coding variants in GWAS data. A more balanced interpretation is necessary.

4. Replicating the findings in an independent dataset would significantly enhance the robustness of the results.

5. The most critical next step is to perform functional studies (in vitro or in vivo) to investigate the proposed mechanisms and establsih a casual link between the identified genes and IS risk.

6. Future research should include data from diverse populations to account for potential ethnic variations in genetic predisposition.

7. Replicating the analyses in independent datasets with similar or different ancestries is vital for confirming the associations.

8. Exploring other omics data, such as gene expression, proteomics, and metabolomics, could provide further insights into the biological pathways linking the identified genes to IS.

Additional comments

1. The paper could benefit from further clarification on how TabNet's attention scores are interpreted, particularly in the context of their use for SNP ranking. A brief explanation of TabNet's underlying mechanism and how its "explainability" feature is leveraged for this purpose would add value.

2. The paper should highlight the potential of this approach for other complex diseases with polygenic inheritance, encouraging its broader application in future GWAS analyses.

3. The paper should discuss any inherent limitations of the utilized data (e.g., the additive model assumption) and acknowledge how these limitations could impact the generalizability of the findings.

---

## Round 0.3 · accepted · Accept

The authors have carefully considered and responded to all of the reviewers' comments. We believe that the manuscript has been significantly improved as a result of these revisions and is now ready for publication.

Reviewer 1 ·

Basic reporting

All comments have been added in detail to the last section.

Experimental design

All comments have been added in detail to the last section.

Validity of the findings

All comments have been added in detail to the last section.

Additional comments

Review Report for PeerJ Computer Science
(Genome-wide association studies of ischemic stroke based on interpretable machine learning)

Thanks for the revision. Both the relevant changes made to the paper and the responses to the reviewer comments are sufficient. For this reason, I recommend that the paper be accepted. I wish the authors great success in their future projects and papers. Kind regards.

Reviewer 2 ·

Basic reporting

The issues have been improved.

Experimental design

The issues have been improved.

Validity of the findings

The issues have been improved.

Additional comments

The issues have been improved.

Reviewer 3 ·

Basic reporting

The paper cites relevant and fairly up-to-date literature. It successfully covers existing research on ischemic stroke (IS) genetics, the limitations of traditional GWAS, and examples of machine learning (ML) applied to GWAS datasets.

Experimental design

The methods employed for GWAS data analysis are well-established, and their application appears appropriate. The use of cross-validation for model selection, exploring multiple aggregation methods for Pareto optimization, and inclusion of established IS-associated gene sets for comparison enhances the paper’s rigor.

Authors provide code at a Github repository, supporting replication. Using data from the publicly available Database of Genotypes and Phenotypes makes the work readily accessible to others.

Validity of the findings

Using real genotypic data from a well-established database is a strength. The statistical methods for each analysis appear correctly chosen. However, a lack of specifics on the statistical analysis methods makes it difficult to thoroughly assess the findings.

Conclusions are clearly stated and largely limited to supporting data. However, claims like "We believe the common genes for different stages of pathogenesis may exist" could be tempered with evidence or be rephrased as areas needing further research.

Additional comments

The unique use of Pareto optimality is the paper’s most interesting contribution, yet its advantages are not discussed in detail. Authors may expand on why this method enhances robustness compared to single metric approaches.